



# Vortex Particle-Mesh simulations of Vertical Axis Wind Turbine flows: from the blade aerodynamics to the very far wake

Philippe Chatelain[1], Matthieu Duponcheel[1], Denis-Gabriel Caprace[1], Yves Marichal[1,2], and Grégoire Winckelmans[1]

[1]Institute of Mechanics, Materials and Civil Engineering, Université catholique de Louvain, 1348 Louvain-la-Neuve, Belgium
[2]Wake Prediction Technologies (WaPT), Rue Louis de Geer 6, 1348 Louvain-la-Neuve, Belgium

*Correspondence to:* P. Chatelain (philippe.chatelain@uclouvain.be)

**Abstract.** A Vortex Particle-Mesh (VPM) method with immersed lifting lines has been developed and validated. Based on the vorticity-velocity formulation of the Navier-Stokes equations, it combines the advantages of a particle method and of a mesh-based approach. The immersed lifting lines handle the creation of vorticity from the blade elements and its early development. LES of Vertical Axis Wind Turbine (VAWT) flows are performed. The complex wake development is captured in details and over very long distances: from the blades to the near wake coherent vortices, then through the transitional ones to the fully developed turbulent far wake (beyond 10 rotor diameters). The statistics and topology of the mean flow are studied. The computational sizes also allow insights into the detailed unsteady vortex dynamics, including some unexpected topological flow features.

## 1 Introduction

The aerodynamics of Vertical Axis Wind Turbines (VAWTs) are inherently unsteady, which leads to vorticity shedding mechanisms due to both the lift distribution along the blade and its time evolution. This translates into a wake topology that is far more complex and unsteady than for their Horizontal Axis counterparts (HAWTs), a characteristic which could be indicative of more intense wake decay mechanisms for VAWTs. Additionally, their inherent insensitivity to wind direction changes hints at a more robust efficiency in turbulent conditions. Logically, both traits have led to several claims of an advantage of VAWTs over HAWTs in wind farms (e.g., Paquette and Barone, 2012; Kinzel et al., 2012), and thence the promises of higher power extraction densities. Those, together with potential operational gains (maintenance costs, the disappearance of yawing actuation), have spurred some definite research momentum in VAWT aerodynamics, in the shape of experimental (Ferreira et al., 2009; Beaudet, 2014) and numerical (Scheurich, 2011; Ferreira et al., 2014) studies. However, because of their unsteady aerodynamics, VAWT simulation and modeling tools have not reached yet the level of development of those for HAWTs, e.g. the Blade Element Momentum method. Numerical investigations of VAWT wake phenomena have only been tackled recently (Scheurich and Brown, 2013) but the volume of these efforts is quite underwhelming when compared to all the comparable works on HAWTs (Sørensen et al., 2015) and the computational domains and resolutions of existing studies are quite limited. In this paper, we perform large-scale, highly-resolved Large Eddy Simulation of the flows past Vertical Axis Wind Turbines by means of



a state-of-the-art Vortex Particle-Mesh (VPM) method combined with immersed lifting lines (Chatelain et al., 2013). We focus on the intrinsic vortex dynamics and wake decay mechanisms; all simulations are thus carried out without turbulence in the wind. The simulation tool is validated against experimental aerodynamic data and is then run for a standard, medium-solidity, H-shaped machine: mean flow and turbulence statistics are computed over more than 15 diameters downstream of the machine.

The sensitivity of the wake behavior to the operating conditions (Tip Speed Ratio, TSR) and to the machine aspect ratio (AR) is also assessed. This paper is structured as follows. We briefly recall the Vortex Particle-Mesh (VPM) method in Section 2 and present some of the advances that enabled the Large Eddy Simulation of wind turbines within this VPM context: the multiscale Sub-Grid Scale model and the modeling of blades through immersed lifting lines. Section 3 presents some validation of our methodology, then moves to the study of a standard VAWT from the perspectives of its aerodynamics and its wake dynamics.

We close this paper in Section 4 with our conclusions and perspectives.

## 2    Methodology

### 2.1    The Vortex Particle-Mesh method

The coarse scale aerodynamics and the wake of the VAWT are simulated using a massively parallel implementation of a Vortex Particle-Mesh flow solver. The present method relies on the Large Eddy Simulation in the vorticity-velocity formulation for

incompressible flows ($\nabla \cdot \boldsymbol{u} = 0$)

$$\frac{D\boldsymbol{\omega}}{Dt} = (\boldsymbol{\omega} \cdot \nabla)\,\boldsymbol{u} + \nu \nabla^2 \boldsymbol{\omega} + \nabla \cdot \mathbf{T}^M \tag{1}$$

where $\nu$ is the kinematic viscosity, and $\mathbf{T}^M$ is the Sub-Grid Scale (SGS) model. The velocity field is recovered from the vorticity by solving the Poisson equation

$$\nabla^2 \boldsymbol{u} = -\nabla \times \boldsymbol{\omega} \,. \tag{2}$$

The advection of vorticity is handled in a Lagrangian fashion using particles, characterized by a position $\boldsymbol{x}_p$, a volume $V_p$ and a vorticity integral $\boldsymbol{\alpha}_p = \int_{V_p} \boldsymbol{\omega} d\boldsymbol{x}$

$$\frac{d\boldsymbol{x}_p}{dt} = \boldsymbol{u}_p \tag{3}$$

$$\frac{d\boldsymbol{\alpha}_p}{dt} = \left( (\boldsymbol{\omega} \cdot \nabla)\boldsymbol{u} + \nu \nabla^2 \boldsymbol{\omega} + \nabla \cdot \mathbf{T}^M \right)_p V_p \,, \tag{4}$$

where we identify the roles of the velocity field in the advection, and of the vortex stretching, diffusion and SGS terms for the

evolution of vorticity.

The right-hand sides of these equations are evaluated efficiently on an underlying mesh (Chatelain et al., 2008). The stretching and diffusion operators use fourth-order finite differences and Eq. (2) is solved efficiently with a Fourier-based solver. In this work, we rely on the technique used by Chatelain and Koumoutsakos (2010), which handled a combination of periodic and unbounded directions through the approach of Hockney and Eastwood (1988). It is here extended to an inflow-outflow



direction, say $x$, and two unbounded directions, $y$ and $z$. A Fourier transform along $x$ yields

$$\nabla^2_{yz}\tilde{\boldsymbol{u}} - k_x^2\tilde{\boldsymbol{u}} = -\widetilde{\nabla \times \boldsymbol{\omega}} \,, \tag{5}$$

where $\tilde{\boldsymbol{u}}(k_x, y, z)$ stands for the $x$-transformed field. For a given $k_x$ mode, this is a two-dimensional Helmholtz equation in an unbounded $(y, z)$-domain; it is solved through a convolution with the corresponding Green's function in Fourier space through the domain-doubling technique of Hockney and Eastwood (1988), see (Chatelain and Koumoutsakos, 2010) for details. The wavenumbers $k_x$ are here constrained to produce inflow-outflow conditions, or equivalently to only permit adequately-phased sine or cosine modes. The following conditions on the streamwise velocity are then imposed: $u_x(0, y, z) = U_\infty$ at the inflow, and $\frac{\partial u_x}{\partial x}(L_x, y, z) = 0$ at the outflow. These are completed with the conditions on the transverse components $\frac{\partial u_y}{\partial x}(0, y, z) = \frac{\partial u_z}{\partial x}(0, y, z) = 0$ and $u_y(L_x, y, z) = u_z(L_x, y, z) = 0$.

The SGS model is a simplified version of the Variational Multiscale (VM) model (Hughes et al., 2001), known as the Regularized version (RVM) (Jeanmart and Winckelmans, 2007). In that variant, the SGS model is designed as an eddy viscosity model acting only on the small scale field

$$\mathbf{T}^M = \nu_{SGS}\left(\nabla\boldsymbol{\omega}^s + \nabla^t\boldsymbol{\omega}^s\right) \tag{6}$$

where $\boldsymbol{\omega}^s$ represents the small-scales part of the vorticity field obtained by high-pass filtering. The eddy viscosity is taken as $\nu_{SGS} = C_r^{(n)}\Delta^2\left(2\mathbf{S} : \mathbf{S}\right)^{1/2}$ where the strain rate $\mathbf{S}$ is evaluated using the complete velocity field. We refer to Cocle et al. (2007, 2008, 2009) for implementation details and for the values of the coefficients $C_r^{(n)}$ when using filtering of order $2n$.

In order to carry out the computational steps above, information is made available on the mesh, and recuperated from the mesh, by interpolating back and forth between the particles and the grid using high order interpolation schemes. Advantageously, this hybridization does not affect the good numerical accuracy (in terms of diffusion and dispersion errors) and the stability properties of a particle method. The present method indeed still waives the typical CFL constraint for the explicit time integration of advection, $\Delta t < C_u h / \|\boldsymbol{u}\|_{\max}$, and rather involves higher order constraints (Koumoutsakos, 2005), e.g. $\Delta t < C_{\nabla u} / \|\nabla\boldsymbol{u}\|_{\max}$; this essentially corresponds to preventing particle trajectories from crossing each other. The time integration scheme used in the present work is a low-storage third order Runge-Kutta (Williamson, 1980).

This last discussion actually pertains to the issue of Lagrangian distortion in particle methods. If left alone, particles can be seen to deplete regions of the flow or cluster in others. Several remedies have been proposed. Dissipative terms can be added to the particle ODEs in order to limit the particle deformations (Monaghan, 2000); this comes at the price of artificial bulk and shear viscosities. State-of-the-art particle methods, such as the present one, rely on a procedure called remeshing (Cottet, 1996; Koumoutsakos, 1997; Ploumhans and Winckelmans, 2000; Winckelmans, 2004), which consists in the periodic regularization of the particle set onto a mesh. This procedure typically relies on high order interpolation formulas (Monaghan, 1985; van Rees et al., 2011) which do involve well controlled levels of artificial viscosity. All the simulations of Section 3 have involved a remeshing operation every 5 time steps that uses the third order accurate $M_4'$ scheme of Monaghan (1985); the same scheme is used for the particle-to-mesh and mesh-to-particle interpolation operations.



## 2.2 Immersed lifting lines

The generation of vorticity along the blades is accounted for through an immersed lifting line approach (Chatelain et al., 2013). The approach is very much akin to a Vortex Lattice method and relies on the Kutta-Joukowski theorem (see e.g. (Prandtl, 1923)) that relates the developed lift $\boldsymbol{L}$ to the relative flow $\boldsymbol{V}_{\mathrm{rel}}$ and the circulation bound around the local 2D airfoil

$$\boldsymbol{L} = \rho \boldsymbol{V}_{\mathrm{rel}} \times \boldsymbol{\Gamma} \,. \tag{7}$$

Lift can also be obtained from the relative flow, its angle of attack $\alpha$ and the airfoil lift coefficient $C_l(\alpha)$; equating this aerodynamics-provided expression to Eq. (7) allows to solve for the instantaneous circulation $\boldsymbol{\Gamma}$ at a blade location. The solenoidal property of vorticity then imposes that streamwise and spanwise vorticities be shed from the lifting line in order to account for spanwise and temporal variations of $\boldsymbol{\Gamma}$, respectively. Over a time step, the shed vorticity is constructed thanks to

Lagrangian tracers released along the blade. The vorticity bound to the blade and the newly generated vorticity are discretized by means of particles immersed in the mesh and in the bulk flow-representing particles; their treatment thus fits within the present particle-mesh framework. Unlike the mesh-only Vorticity Transport Model (Brown and Line, 2002) or an Actuator Line technique (Sørensen et al., 2015), this treatment of vorticity sources is Lagrangian and well suited for the large time steps enabled by the rest of the method. The aerodynamic behavior of the lifting lines sections, i.e. $C_l$ and $\boldsymbol{\Gamma}$, account for unsteady

effects through a Leishman-Beddoes dynamic stall model (Leishman, 2006). This semi-empirical model shows a good trade-off between simplicity and accuracy, provided that the model coefficients are validated with relevant experimental data. In this work, we follow the indications of Dyachuk (Dyachuk et al., 2014) and Scheurich (Scheurich, 2011) who present coefficients for various airfoils validated in the particular case of a VAWT.

## 3 Results

### 3.1 Validation

We first present validation results against recent work (Castelein, 2015) for a low solidity two-bladed H-shaped machine with NACA0018 airfoils. The parameters for the Leishman-Beddoes dynamic stall model are based on those for a NACA0015 in (Scheurich, 2011); they are here tuned to fit the static behavior of the polar at the Reynolds number of the experiment at the design point, $Re = U_{\mathrm{rel}} c / \nu = 1.5\,10^5$. Throughout this paper, we use the following axes convention: $x$ is the streamwise

direction, $y$ is cross-stream and orthogonal to the VAWT axis, which $z$ is parallel to. The origin for the blade angular position $\theta$ is set at the upwind-going position. Figure 1 presents the profiles of the normal and tangential forces developed by a blade over a revolution, non-dimensionalized with respect to the profile chord $c$ and the dynamic pressure $q_0 = 1/2\,\rho U_\infty^2$. While the results at an intermediate TSR $= \Omega R / U_\infty$ show good agreement (Fig. 1(b)), there is a clear departure at the lower TSR (Fig. 1(a)). The experimental points hint at a stall happening later on the upstream stretch, around $90°$, and more abruptly than

for the simulation. This mismatch on the upstream part has a direct influence on the predictions for the downstream stretch ($\theta \in [210°, 270°]$), as the stall-generated structures are advected through the rotor; this may explain the marked differences





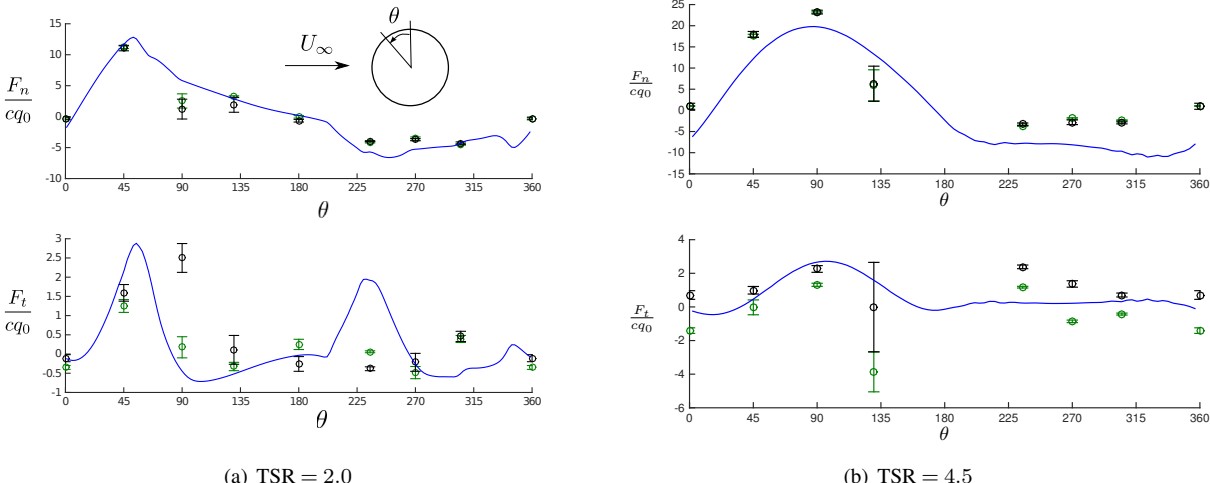

(a) TSR = 2.0              (b) TSR = 4.5

**Figure 1.** Validation: evolution of the normal, $F_n/(cq_0)$, and tangential, $F_t/(cq_0)$, force coefficients at mid-height vs the blade angular position $\theta$; VPM simulation (solid line), experimental results ($o$) with two techniques of force computation from PIV flow fields (Castelein, 2015).

observed there. These results are satisfactory: they are indeed very sensitive to the dynamic stall model, here probably still misadapted, and to some unquantified uncertainties for the experimental facility (the TU Delft Open Jet Wind Tunnel), namely its blockage and secondary flows in the test section.

### 3.2 Aerodynamics

The remainder of this section focuses on a low solidity H-VAWT studied numerically by Scheurich et al. (2010); Scheurich (2011). For the sake of completeness, we here briefly recall its main parameters: an aspect ratio $AR = H/D = 1.5$, a solidity $\sigma = nc/D = 0.1725$, and constant-chord NACA0015 airfoils. Figure 2 shows this machine's power coefficient ($C_p$) curve as a function of the TSR. In order to be computationally affordable, the whole curve has been produced using an intermediate resolution of $D/h = 48$ mesh points/particles per diameter; it allows to identify the optimum power operating point at TSR =
3.21. We investigate the behaviors of the aerodynamics and wake topology of this baseline point, two off-design points (TSR = 2.14, 4.28), and also different aspect ratios $AR = 1.0, 3.0$. These configurations have been simulated at a fine resolution $D/h = 96$ and in domains that extended up to 17 diameters downstream of the rotor axis.

Figure 3 presents the aerodynamic behavior at mid-height. A lower resolution result ($D/h = 48$) is also shown for the baseline TSR and demonstrates the converged state of our simulations. The aspect ratio only affects marginally the aerodynamics
in the middle of the blades; its effects will be discussed further below. A positive angle of attack corresponds to a relative velocity coming from outside of the cylinder swept by the blades. The angle of attack evolution during a revolution is not symmetrical for the upstream and downstream legs because of the reduced velocity encountered downstream. At the baseline




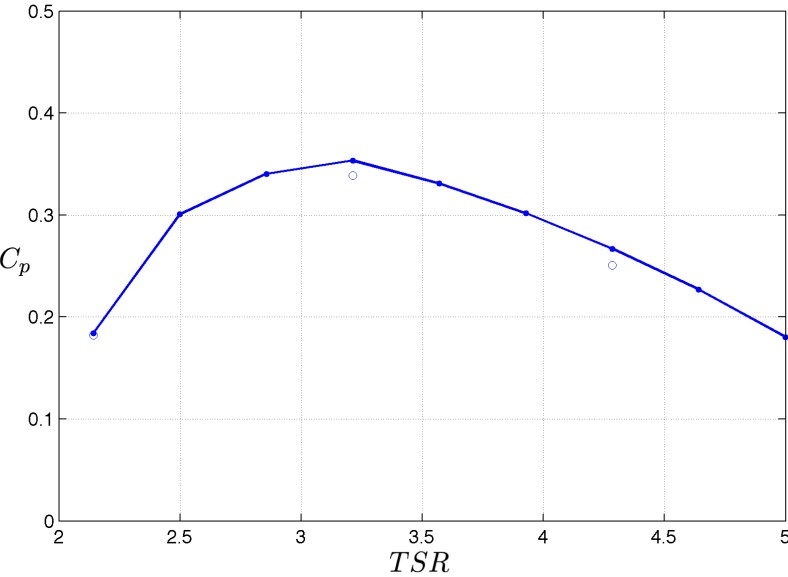

**Figure 2.** H-type VAWT with $AR = 1.5$: power coefficient curve obtained at intermediate resolution ($D/h = 48$, solid line) and configurations investigated at high resolution ($D/h = 96$, circles).

TSR, it reaches a maximum just after the most upstream position ($15°$ around $\theta = 120°$) and the downstream region is characterized by a plateau close to $-7°$. Also of note are the oscillations around $210°$ and $330°$, in the angle of attack and the force coefficients. These are quite well-resolved and physical: as discussed in Section 3.3, the vortex sheets shed during the upstream leg indeed impinge upon the blade in its downstream leg; the velocity jumps associated with these sheets then cause variations in the velocity relative to the blade. The off-design operating points exhibit the expected behaviors: a high TSR will lead to smaller angles of attack and a decreased torque production while the low TSR causes a distinctive stall in the upstream region and also in the downstream one. It is visible in the sharp transitions of the force coefficients at $90°$ and $270°$. The AoA exhibits different behaviors, but consistent with the physics. In the upstream region, the flow is dominated by the blockage effect: as the loading decreases because of stall, the AoA increases even faster; downstream, the blade initially sees a flow less impacted by the stalled upstream part but then encounters the wake of the unstalled part ($\theta \in [0°, 90°]$) and drops rapidly ($\theta = 270°$). Finally, we summarize the effects of TSR, AR and simulation resolution on the estimation of global performance figures in Table 1. As expected, the power, thrust and sideforce coefficients are quite sensitive to the TSR. The machine aspect ratio, however, does not seem to have a major impact on them: going from $AR = 1$ to 3 only improves the $C_P$ by less than 2%.



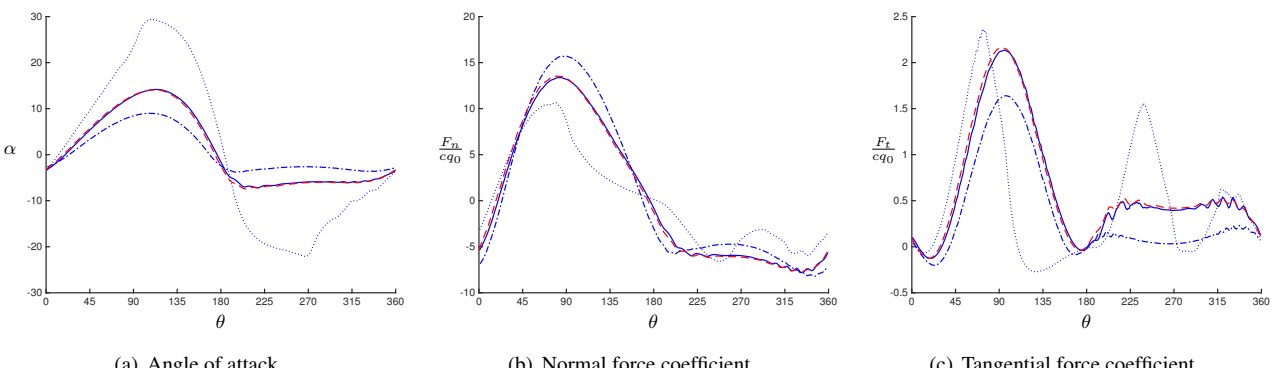

| (a) Angle of attack | (b) Normal force coefficient | (c) Tangential force coefficient |

**Figure 3.** H-type VAWT with $AR = 1.5$: evolution of the angle of attack and of the normal, $F_n/(cq_0)$, and tangential, $F_t/(cq_0)$, force coefficients at mid-span versus the blade angular position $\theta$ at $TSR = 2.14$ (dotted), 3.21 (solid), and 4.28 (dash-dotted); an intermediate resolution ($D/h = 48$) result for $TSR = 3.21$ is also shown (dash).

**Table 1.** H-VAWT global performance: effects of aspect ratio, TSR and spatial resolution.

| AR | TSR | $C_P$ | | $C_x$ | | $C_y$ | |
|---|---|---|---|---|---|---|---|
| | $D/h$ | 48 | 96 | 48 | 96 | 48 | 96 |
| 1.0 | 3.21 | | 0.338 | | 0.844 | | 0.0344 |
| 1.5 | 2.14 | 0.184 | 0.182 | 0.556 | 0.557 | $-0.0518$ | $-0.0376$ |
| 1.5 | 3.21 | 0.353 | 0.339 | 0.863 | 0.845 | 0.0193 | 0.0435 |
| 1.5 | 4.28 | 0.267 | 0.250 | 0.910 | 0.887 | 0.0390 | 0.0683 |
| 3.0 | 3.21 | | 0.344 | | 0.852 | | 0.0616 |

## 3.3 Wakes

### 3.3.1 Vortex dynamics

The instantaneous wakes of the $AR = 1.5$ machine at the three considered TSRs are visualized through volume rendering of the vorticity magnitude in Fig. 4. They allow several insights into the complex vortical structure of the wake which is significantly different from that of a HAWT. We first consider the design TSR (Fig. 4(b)). The vorticity shed in the wake consists in (i) the blade tip vortices, which constitute the top and bottom sides of the wake, and (ii) the vortex sheets, shed due to the time-variation of the circulation of the blades, which form the lateral sides. The tip vortices are the strongest in the vicinity of the upstream- and downstream-most positions of the blades ($\theta = 90°$ and $270°$) where the blades operate at their maximum angle of attack. There, depending on the appearance of stall, or delayed stall effects, the blade will achieve its maximum





(a) TSR = 2.14

(b) TSR = 3.21

(c) TSR = 4.28

**Figure 4.** H-type VAWT with $AR = 1.5$: volume rendering of the vorticity magnitude $\|\boldsymbol{\omega}\|$; the lifting lines are also shown as 3-D blades.





circulation then lose it either abruptly or progressively, depending on whether the blade is stalled or not. At the design TSR, the blade exploits the delayed stall at its most: its circulation keeps increasing, well past $\theta = 90°$, and then smoothly decreases. The $d\Gamma/dt$ vorticity shedding is maximum when the blades are close to their lateral positions $\theta = 0°$ (upwind leg) and $180°$ (downwind leg). The corners of the wake, i.e. the intersections of the two types of vortical structures described above, give rise

to the fastest-growing vortical instabilities which quickly propagate and cause the pairing of vortices of unequal circulations. Indeed, the unsteady aerodynamics have produced vortices with a varying circulation and the shed vortices will interact with a different section of a preceding/succeeding vortex. In this kind of event, the stronger vortex distorts the weaker one, leading to intense stretching, enstrophy production, and the propagation of disturbances along the vortex cores. This mechanism, most visible in Fig. 4(a), is well known in vortex dynamics and had already been identified on aircraft wakes (Bristol et al., 2004;

Leweke et al., 2016). As a direct consequence, the turbulent regions of the wake grow from the corners and the wake only reaches a fully turbulent state once these regions have merged: the distance to reach this state will be governed directly by the aspect ratio of the machine.

The VAWT wake decay is of course also governed by the TSR in a fashion very similar to that of the HAWT: a high TSR (Fig. 4(c)) induces narrower vortex separations, which directly condition the growth rate of the instabilities and the time to

the reconnection events. This directly, and very geometrically, translates in an increasing opening angle for the envelopes of the corner vortical structures, going from Fig. 4(a) to 4(c)). Decreasing the TSR below the design point actually affects the wake even more dramatically. The stall event on the upstream part of the revolution weakens the upstream wake contribution (between $\theta = 90°$ and $180°$), generating a stopping vortex that will be advected through the rotor (Fig. 4(a)). One can thus expect a two-lobed wake. Conversely, higher TSRs exhibit weaker vortical structures being advected through the rotor. As

discussed in Section 3.2, the $d\Gamma/dt$ sheets shed on the upstream leg will cross the rotor and impact the blade aerodynamics on the downstream leg, with an extreme case being the above-discussed stall event.

The behaviors of the upstream tip vortices within the rotor are more complex to apprehend, as they are affected by several factors: the intrinsic roll-up dynamics of a vortex sheet (with a time-varying strength) and the velocities induced by the surrounding vortical structures, including the bound vortices on the blades. To some degree, the latter can be crudely linked to

the overall rotor loading (the $C_x$ values of Table 1). For a highly loaded rotor (TSR $= 3.21$ and $4.28$), the generated blockage effects will push the vortices shed upstream vertically and away from the downstream blade tips. One only sees the upstream tip vortices impinging upon the downstream blades at a low rotor loading, as it is the case for TSR $= 2.14$. This observation does not agree with the results of Scheurich and Brown (Scheurich and Brown, 2013), which showed upstream vortices colliding the blades at high TSRs. A possible explanation might lie in the relatively short domain and the direct use of the unbounded

Biot-Savart law in their work. One needs to add additional terms to enforce an outflow condition for this otherwise clipped vorticity field; the present study does precisely that by enforcing a normal outflow velocity ($\partial u/\partial x = 0$, $v = 0$, $w = 0$) through its Fourier-based solver (Chatelain and Koumoutsakos, 2010). Blockage is but one, and global, factor however. The discussion can be refined as additional, and less immediate, effects are to be expected from the machine geometry. The aspect ratio, as indicated by $C_x$ in Table 1, has a small effect on the blockage and one can also expect an influence on the 3D topology of this

blockage effect: a higher $AR$ thus leads to an increased clearance between the vortices and the blade, as shown in Fig. 5. The





(a) $AR = 1.0$

(b) $AR = 1.5$

(c) $AR = 3.0$

**Figure 5.** Effect of aspect ratio at TSR $= 3.21$: contours of the instantaneous cross-stream vorticity component $\omega_y D/U_\infty$.

number of blades also has a strong influence; the two-bladed machine of Section 3.1 (not shown here, also see (He, 2013)) exhibits such vortex-blade collisions, in spite of its high loading $C_x = 0.874$. Finally, beyond the rotor, the instantaneous vorticity fields of Fig. 5 also offer some insights into the pairing phenomenon of the tip vortices, the generation of a turbulent wake and the recirculation region.

5  ### 3.3.2 Average flow statistics

The average behavior of these wakes is studied through the mean axial velocity $\bar{u}$ and the turbulent kinetic energy $\bar{k}$; these statistics were collected over a period $T_{\mathrm{avg}} = 30\,D/U_\infty$. Figure 6 shows a horizontal slice of these statistics for the $AR = 1.5$ machine. This averaged wake exhibits several prominent features that reflect the phenomena identified in the discussion above. In all the conditions, we observe the generation of TKE on the sides of the wake and the associated smearing of the velocity







(a) TSR = 2.14

(b) TSR = 3.21

(c) TSR = 4.28

**Figure 6.** H-type VAWT with $AR = 1.5$: mean streamwise velocity $\bar{u}/U_\infty$ and turbulent kinetic energy $\bar{k} = \frac{\overline{u'u'} + \overline{v'v'} + \overline{w'w'}}{2U_\infty^2}$ in the $y/D = 0$ plane.





deficit. This is consistent with our discussion of the vortical instabilities in the corner structures and the subsequent propagation of the turbulent regions. At low TSR, the averaged velocity field does exhibit the expected two-lobed structure, with a stronger deficit on the blade-travelling-upwind side of the rotor ($\theta \in [270°, 90°]$). For the higher TSRs (Figs. 6(b) and 6(c)), a backflow region lies inside the wake at a position that varies with the TSR: it is centered at $x/D \simeq 5.5$ for $TSR = 3.21$ and at $x/D \simeq 4$

for $TSR = 4.28$. The location of this feature clearly coincides with the production of TKE and an accelerated smearing of the wake velocity deficit; this too agrees with our vortex dynamics discussion. The topology of the associated recirculation bubbles is clearly three-dimensional and will not be discussed here.

Finally, the averaged wakes exhibit a slight deviation in this mid-plane. As expected, the behaviors of the three TSRs do correlate with the signs and values of the side-forces produced by the rotor (see $C_y$ in Table 1). These side-forces also appear

in the average behavior as observed in cross-flow slices (Fig. 7). The deformation of the velocity deficit clearly hints at the presence of mean streamwise vortices along the corners of the wake (Figs. 7(a) and 7(b)), a clear departure from a HAWT wake.

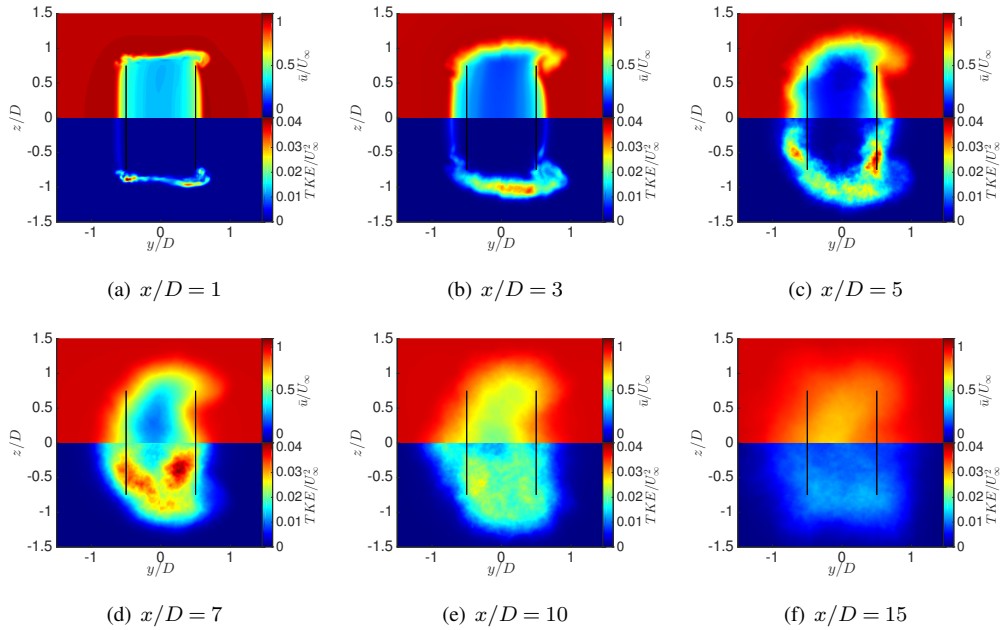

**Figure 7.** H-type VAWT with $AR = 1.5$: mean streamwise velocity $\bar{u}/U_\infty$ and turbulent kinetic energy $\bar{k}$ in cross-flow slices.

### 3.3.3 Decay diagnostics

We apply classical turbulent wake diagnostics to the characterization of the wake decay. More specifically, we adapt integral

quantities, such as the displacement and momentum widths, to the present context; the wakes being considered indeed lack symmetry and exhibit strong secondary flow structures, which makes the use of a pointwise velocity deficit unsuitable. Thus





we define dimensionless displacement and momentum surfaces, respectively as

$$S_1(x) \;=\; \frac{1}{HD} \int\limits_{-\infty}^{\infty} \int\limits_{-\infty}^{\infty} \left(1 - \frac{u_x(x,y,z)}{U_\infty}\right) dy\,dz \qquad (8)$$

$$S_2(x) \;=\; \frac{2}{HD} \int\limits_{-\infty}^{\infty} \int\limits_{-\infty}^{\infty} \left(1 - \frac{u_x(x,y,z)}{U_\infty}\right)\left(\frac{u_x(x,y,z)}{U_\infty}\right) dy\,dz \;. \qquad (9)$$

These diagnostics correspond to integrals of flux quantities in cross-stream sections located at a distance $x$ downstream of the turbine axis. $S_1$ quantifies the blockage effect caused by the wake on the flow; it is shown in Fig. 8(a). As a reference, $S_1$ should be compared with the square of the displacement width ($\delta^2$) of an axisymmetric wake for which classical similarity theory (Tennekes and Lumley, 1972) predicts a behavior $S_1 \sim x^{-1/3}$ in the far wake. The asymmetry of the wake generator and its proximity are such that we cannot observe the self-similarity region: classical results for bluff bodies indicate a development distance of $x/D \sim 50$ to obtain the theoretical far wake self-similarity. The decay observed for $x/D > 5$ for most of the configurations does however hint at a power-law-like behavior; its thorough analysis over even greater distances goes beyond the scope of the present work. Still, the evolution of $S_1$ does provide a signature of the recirculation region: the magnitude and the extent of the overshoot $S_1 > 1$ correlates with the location and the size of the recirculation bubble for the design and high TSRs (Figs. 6(b) and 6(c)). This correspondence also agrees with the effect of the aspect ratio: an increasing AR pushes both the recirculation (indicated by the merging of vortical structures in the center of the wake in Fig. 5) and the $S_1$ overshoot further downstream.

The dimensionless momentum surface $S_2$ is related to the deficit in the flux of momentum in these planes. In the absence of secondary flows and pressure gradients, it should in fact correspond to the thrust coefficient $S_2 \simeq C_x$ at large distances behind the VAWT when a factor 2 is used in the definition of $S_2$, as here in Eq. (9). This is confirmed by our results of Fig. 8(b)): after a transition, the curves tend towards the corresponding $C_x$ values of Table 1.

Finally, the case TSR $= 2.14$ constitutes an outlier in the discussions above. This is not unexpected: the instability growth is slower than for the other cases and does not allow the transition to a well-mixed fully-turbulent wake within the computational domain.

## 4 Conclusions

A Vortex Particle-Mesh method, here briefly presented, has been applied to large scale and high resolution LES of VAWT wakes. The method is capable of tracking vortical structures over very long times and distances. This has led to several insights into the vortex dynamics at work inside the wakes of VAWTs. The mean flow topology has been extracted; unsteady flow aspects, three-dimensional effects and classical wake diagnostics have also been studied. The impact of several of these flow features for the deployment of VAWTs in wind farms is considerable: the aspect ratio and the operating conditions of the machine greatly affect the wake decay, and even allow the presence of a recirculation region. The present study merely constitutes a preliminary study of VAWT wakes. Direct follow-up work will investigate the 3D topology of the averaged wake




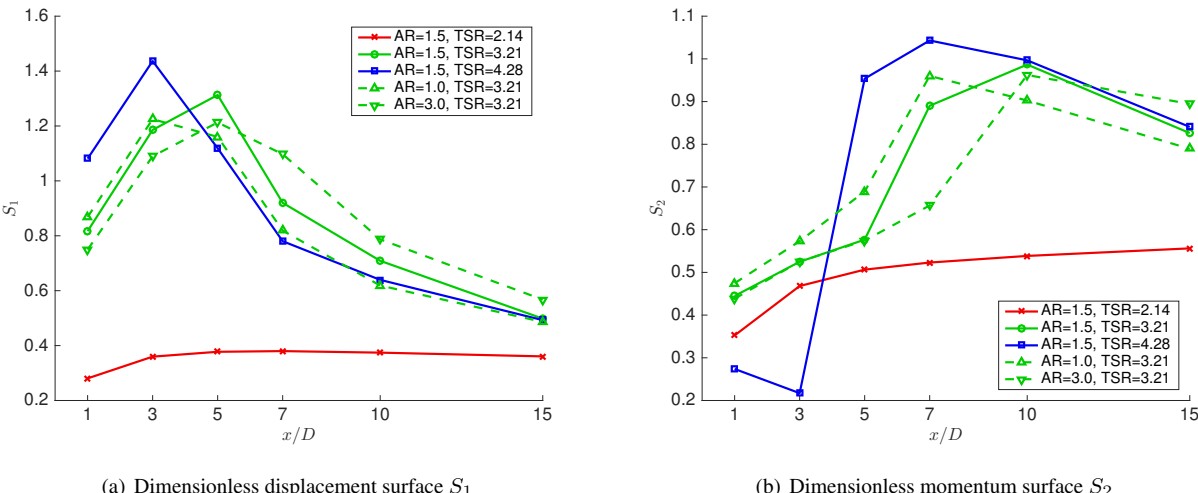

(a) Dimensionless displacement surface $S_1$      (b) Dimensionless momentum surface $S_2$

**Figure 8.** H-type VAWT: dimensionless displacement and momentum surfaces as functions of the streamwise coordinate.

and its unsteadiness. We will then also consider the behavior of these machines and of their wakes in a turbulent wind. Our methodology can also accommodate rotor dynamics models and realistic controllers; this will bring definitive answers to the smoothness of torque generation for H-type VAWTs and their performances in wind farms.

## 5 Code availability

5    The Immersed Lifting Line VPM code and its Fourier-based solver library are proprietary. The Parallel Particle-Mesh (PPM) library is an open source library (ETHZ/CSE Lab, 2011).

## 6 Data availability

The data sets involved in this study consist of massive 3D and time-dependent data sets, the handling of which is not tractable on a data registry.

10    *Author contributions.* P. Chatelain and M. Duponcheel prepared and ran the simulations and D.-G. Caprace performed their post-processing. P. Chatelain and M. Duponcheel developed the code; Y. Marichal and D.G. Caprace developed the dynamic stall model inside the code. P. Chatelain, M. Duponcheel and G. Winckelmans contributed to the analysis and the discussion of the results. P. Chatelain prepared the manuscript with contributions from all co-authors.

*Competing interests.* The authors declare that they have no conflict of interest.



*Acknowledgements.* The authors acknowledge the fruitful discussions with Thierry Maeder, Stefan Kern and Dominic von Terzi, at the Aerodynamics and Acoustic Lab at GE Global Research, Garching bei München. Matthieu Duponcheel was partially supported by the ENGIE-funded research project *Small Wind Turbines*. The development work benefited from the computational resources provided by the supercomputing facilities of the Université catholique de Louvain (CISM/UCL) and the Consortium des Équipements de Calcul Intensif

5    (CÉCI) en Fédération Wallonie Bruxelles (FWB) funded by the Fond de la Recherche Scientifique de Belgique (F.R.S.-FNRS) under Convention No. 2.5020.11. The production simulations used computational resources made available on the Tier-1 supercomputer of the FWB, infrastructure funded by the Walloon Region under the Grant Agreement No. 1117545.





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
