# Peer review of "Vortex Particle-Mesh simulations of Vertical Axis Wind Turbine flows: from the airfoil performance to the very far wake"

_Wind Energy Science, 2016_

## Referee Comment (RC1) · Anonymous Referee #1 · 1 Mar 2017

This manuscript is well-prepared, with a good overview of the scientific problem, adequate description of the methods used, presentation of verification and validation evidence for simulation results, and a good description of physical observations.

The following recommendations are made to improve the quality of the paper. 1. In Figure 1, please provide some explanation for the discrepancy in the two experimental measurements of tangential force coefficient near theta=90 degrees for tsr=2 and near 135 degrees for tsr=4.5. In the prior case, the error bars in the measurements do not overlap, which casts doubt on these data and confused at least this reader. 2. Please provide the Reynolds number for the VAWT flow that is being studied. There should also be some discussion on the state of the initial shear layers and tip vortices. Are

these expected to be initially laminar or turbulent, based on the flow Reynolds number? Is the SFS model active in the early wake shear layers/vortices? Are these initial flow structures in a transitional regime, and if so, how much confidence do you have in the ability of the LES model to correctly predict transition from an initially laminar to turbulent state? Might this affect some of the behavior of the wake instability and subsequent breakdown? 3. Related to the previous point, does Figure 6 show the resolved TKE, or total TKE (resolved+modeled)? 4. In paragraph 10, Page 12, there is a discussion on the possible presence of mean streamwise vortices through investigation of the velocity and TKE fields. Why not look at the streamwise vorticity field directly?

---

## Referee Comment (RC2) · Anonymous Referee #2 · 7 Mar 2017

The paper is well written and presents a high quality work, that is both interesting and relevant.

Title: the title of the paper as a sub-title: "from the blade aerodynamics to the very far wake". The actuator/lifting line method does not present detail at chord level; therefore, although the model is suitable for blade scale aerodynamics, the current formulation of the title is not accurate. The analysis is limited to eight diameters downstream, for which the reference to "very far wake" is not accurate. I suggest revising the title.

Abstract: the authors mention "very long distances". I suggest a more precise characterization, as for example, "up to 8 diameters downstream".

[Figure]

Comments in the body of the paper:

1. P5, fig 1 and its discussion, it is stated that for TSR=4.5, the results compare well with experiments, although Fn is clearly underpredicted.

2. P7, the maximum angle of attack are not at \theta=90 and 270 degrees

3. P9, "At the design TSR,the blade exploits the delayed stall at its most:" – please explain what this means

4. P9, "this mechanism, most visible in fig 4a, is well known in vortex dynamics and had already been identified on aircraft wakes." I suggest a figure where you highlight this event. It is a too complex process for a reader to follow form this short description.

5. P9, "One needs to add additional terms to enforce an outflow conditions for this otherwise clipped vorticity field; the present study does precisely that by enforcing a normal outflow velocity through its Fourier-based solver." This explanation is not clear. a. The comparison with the work of Scheurich uses TSR as a term of comparison; wouldn't loading be a more relevant term? The strength of the tip vortex is dependent also on the airfoil used; are these comparable? b. Other authors have used free wake vortex filament models, and have seen the same effect of inboard motion. How does this hold with the suggestion that a term should be added to the FFT solution in a meshed domain?

6. P10, "The number of blades also has a strong influence; the two-bladed machine of Section 3.1 exhibits such vortex-blade collisions, in spite of its high loading." How does this relationship work?

7. P11, For the discussion about figure 6, to get a better insight into the mechanism of turbulence creation in the wake corners, would it maybe be valuable to add an impression from the side as well? Then it is easier to see what happens in the corners.

8. P12, "The deformation of the velocity deficit clearly hints at the presence of mean streamwise vortices along the corners of the wake, a clear departure from a HAWT

wake." This is only true for a HAWT subject to axial flow, but certainly not for inclined inflow for which a lateral force component is apparent (which is in practice always the case).

9. P12, "..., which makes the use of a pointwise velocity deficit unsuitable." - suggestion -Which makes the definition of a velocity deficit profile based upon a single characteristic point unsuitable.

10. P13, The value of S1, due to continuity equation, should always be zero, except for the fact that Uinf is corrected to the local velocity outside of the wake, which is larger than Uinf. Please explain the use of S1, and its modified application.

11. P13, maybe add a reference for the x/D~50 statement? And probably a discussion about the decay laws for HAWTS, in practise these seem to deviate significantly from bluff-body flow rules in case of turbulent inflow.

———————————————

---

## Author Comment (AC1) · 4 Apr 2017

**Vortex Particle-Mesh simulations of Vertical Axis Wind Turbine flows: from the blade aerodynamics to the very far wake**
**Reply to reviewers**

P Chatelain[a], M Duponcheel[a], D-G Caprace[a], Yves Marichal[b], G Winckelmans[a]

[a]*Institute of Mechanics, Materials and Civil Engineering, Université catholique de Louvain, 1348 Louvain-la-Neuve, Belgium*
[b]*Wake Prediction Technologies (WaPT), Rue Louis de Geer 6, 1348 Louvain-la-Neuve, Belgium*

**1. Introduction**

First and foremost, the authors wish to thank the reviewers for their time, their careful review, and their appreciation of the work. We are most grateful for the insightful comments, which we have addressed thoroughly below and in the revised paper. We hope this will satisfy the reviewers.

**2. Reviewer 1**

Below we reply to every point brought by this reviewer. The corresponding changes are highlighted in blue in the revised manuscript; changes that address a comment common to both reviewers are shown in orange.
* * *
**Comment 1.** In Figure 1, please provide some explanation for the discrepancy in the two experimental measurements of tangential force coefficient near theta=90 degrees for tsr=2 and near 135 degrees for tsr=4.5. In the prior case, the error bars in the measurements do not overlap, which casts doubt on these data and confused at least this reader.
* * *
As stated by the author of the experiment in their conclusions [1]: "the load determination method is unreliable at this position, since the blade is in deep stall. The momentum was not conserved, what creates large variations on the loads for the different contours, resulting in an error on the mean load value. This azimuth position can not be used for comparisons in tangential direction." However, we agree with the reviewer that the non overlapping, large error bars are confusing, and a comment on the validity of experimental data at these specific angular position could be done. The following modification will be brought to the text: "The experimental points hint at a stall happening later on the upstream stretch, around 90°, and more abruptly than for the simulation; we report here that the authors of the experiment advised to use circumspection for the $F_n$ data at 135° and clearly question the validity of their results for $F_t$ through the whole rotation. We nevertheless confront our simulations to all their results in Fig. 1."

———————————————

**Comment 2.** Please provide the Reynolds number for the VAWT flow that is being studied. There should also be some discussion on the state of the initial shear layers and tip vortices. Are these expected to be initially laminar or turbulent, based on the flow Reynolds number? Is the SFS model active in the early wake shear layers/vortices? Are these initial flow structures in a transitional regime, and if so, how much confidence do you have in the ability of the LES model to correctly predict transition from an initially laminar to turbulent state? Might this affect some of the behavior of the wake instability and subsequent breakdown?

——————————

We agree with the reviewer to say that the original text contained only little information about the transition to turbulence.

- The Reynolds number based on the chord of the blades and on the tangential velocity $U = \omega R$ amounts to $Re = 4.0\,10^5$. In that regime, we would

expect the boundary layers to be transitional or turbulent, and thus the shear layers to rapidly grow turbulent downstream of the blades. The resolution of the current simulations and the current implementation of the lifting line in the Vortex Particle Mesh method do not allow to capture fine scale perturbations within the shed vortical structures; only the "larger scales" of the shed vortex sheets are captured, in the form of spanwise and streamwise vorticity components due to the spatial and temporal variations of lift. These vortex sheets are thus generated in a laminar way; they are however sensitive to core size instabilities: starting or end effects due to temporal variations, internal waves (Kelvin modes),...

We have added a mention of the blade Reynolds number and a brief contextual remark about the limitations of the lifting line approach in the section concerning the lifting lines: "We note that all these methods are not able to capture the sub grid scale structure of the actually shed structures."

- Our Sub-Grid Scale (SGS) model is actually acting on the small resolved scales of the flow following the *complete-small* Reduced Variational Model (RVM) implementation. This means that turbulence modeling is acting on the shed structures directly behind the blade, albeit in a very controlled fashion, and thus does not dissipate them too quickly.

- Although the initial flow structure are shed in a laminar like manner, we do believe that our simulations correctly capture the transition to a fully developed turbulent wake. The above-mentioned lacking internal perturbations have no effect on the large-scale trajectories of the vortices, which are precisely governing the vortex-vortex interactions and reconnections. As they generally occur between vortices of different intensities, these events are very intense and produce a great amount of stretching, actually overwhelming the generation of small scales and turbulence.

As the interactions are more frequent at the corner of the wake, the transition to turbulence is propagating from those regions to the rest of the

wake, what we meant to say when writing: "As a direct consequence, the turbulent regions of the wake grow from the corners and the wake only reaches a fully turbulent state once these regions have merged." We propose the following modification to clarify the relation between vortex interactions and transition: "In this kind of event, the stronger vortex distorts the weaker one, leading to intense stretching, enstrophy production, and the propagation of disturbances along the vortex cores therefore bringing an overwhelming contribution to the transition to turbulence." See also answer to comment 6 of Reviewer 2.

In conclusion, we do not think that the absence of an accurate prediction of the shear layer transition will affect the behavior of wake decay. The wake instabilities, and more generally the wake dynamics, are governed by the circulation of the vortices which we do capture. This dictates the growth rate of instabilities (at least, to the leading order). Nevertheless, we are currently considering the improvement of our lifting line model to also capture the shear layers originating from the blades. A more thorough analysis of the influence of the shear layers on the transition mechanisms will then be possible.

―――――――――――

**Comment 3.** Related to the previous point, does Figure 6 show the resolved TKE, or total TKE (resolved+modeled)?

―――――――

Figure 6 shows resolved TKE only; this is now mentioned in the caption.

―――――――――――

**Comment 4.** In paragraph 10, Page 12, there is a discussion on the possible presence of mean streamwise vortices through investigation of the velocity and TKE fields. Why not look at the streamwise vorticity field directly?

―――――――

Mean streamwise vorticity field is difficult to interpret because of the nearly perfectly periodic behavior of that quantity in the wake. Even with converged

statistics, we observe seemingly noisy patches of axial vorticity of opposite signs, with no clear conclusion to be drawn regarding the dominating streamwise component (see Fig. 1 for example). We now have included such plots and added a short discussion in the text at the end of the discussion of averaged flow quantities: The mean streamwise vorticity at three transverse slices is shown in Fig. X. Even though the statistics are converged, the near-perfect periodicity of the flow leads to a pattern of positive and negative patches, signatures of the advection of tip vortices shed on the upstream and downstream parts of the rotation, respectively. The dominant streamwise vorticity is thus difficult to identify in the near-wake but large scale structures can be identified further downstream, also thanks to the induced deformation of the wake.

[Figure]

(a) $x/D = 1$        (b) $x/D = 5$

(c) $x/D = 10$

Figure 1: H-type VAWT with $AR = 1.5$: mean streamwise vorticity $\bar{\omega}/(U_\infty/L)$.

**3. Reviewer 2**

Below we reply to every point brought by this reviewer. The corresponding changes are highlighted in green in the revised manuscript; changes that addressed a comment common to both reviewers are shown in orange.

———————————————

**Comment 1.** The title of the paper as a sub-title: "from the blade aerodynamics to the very far wake". The actuator/lifting line method does not present detail at chord level; therefore, although the model is suitable for blade scale aerodynamics, the current formulation of the title is not accurate. The analysis is limited to eight diameters downstream, for which the reference to "very far wake" is not accurate. I suggest revising the title.

——————————

We partially agree with the reviewer on that comment. On the first remark, we totally agree: the lifting line model does not represent detailed aerodynamics and therefore, the original title of the manuscript can be misleading. On the second remark, however, we disagree. Our results and analyses entail features up to as far as 15 diameters downstream. Regarding most of today's literature concerning VAWT, this can be considered as "very far wake".

We consider that the formulation "blade scale aerodynamics", suggested by the reviewer, is more suitable to our method; we propose to go even further with the completely unambiguous: "airfoil performance",

Therefore, we would change from the original title to: "Vortex Particle-Mesh simulations of Vertical Axis Wind Turbine flows: from the airfoil performance to the very far wake"

———————————————

**Comment 2.** Abstract: the authors mention very long distances?. I suggest a more precise characterization, as for example, up to 8 diameters downstream?.

——————————

In line with our answer to the previous comment, we consider consider the following modification: "The complex wake development is captured in details and up to 15 diameters downstream: [...]"

———————————————

**Comment 3.** P5, fig 1 and its discussion, it is stated that for TSR=4.5, the results compare well with experiments, although Fn is clearly underpredicted.

——————————

This is true and we are grateful to the reviewer for this remark. The discrepancy is in fact quite visible at high TSR: the machine is then more loaded and the visited angles of attack are in a smaller range than in the low TSR case. We attribute it to the curvature of the relative flow, which is not sensed by the standard lifting line method, but which in reality tends to increase the loading of the blade [2]. In our comparison with Castelein's experiment [1], flow curvature was not taken into account nor modeled. Such a correction is not very complicated to add and it was in fact introduced in our lifting line model in later simulations of the Castelein VAWT. We had not reported on this correction in the original manuscript due to format constraints. The correction, for a VAWT, consists in virtually increasing the angle of attack of the blade by an amount which we compute using the radius of the turbine, the chord of the blade and thin profile theory considerations.

In the new version of the manuscript, we add results with curvature correction in the Validation section, which do confirm our suspicion above and dramatically improves our results. A paragraph has been added in the section devoted to the lifting line: The standard lifting line and the actuator line techniques are not able to capture flow curvature effects. Indeed, if the flow relative to the blade is curved, as it is the case here for a blade in rotation through essentially straight streamlines, the airfoil behaves as an airfoil with an additional camber [3, 4]. We consider a blade with a chord $c$ tangentially positioned at a radius $R$ for its quarter-chord position, this additional camber can be modeled in a straightforward manner by pitching the blade inwards by an angle $\alpha_0 =$

$\arctan\left((1 - \cos(\beta/2))/\sin(\beta/2)\right)$ where $\beta = \arctan(c/2\,R) + \arctan(c/2\,R)$. In the validation section below (Sec 3.1), we verify the positive effect of such a correction.

The following discussion has been added in the validation section: We report on VPM simulations with and without a curvature correction, which here amounts to a inward pitch $\alpha_0 = 1.72°$. This correction appears to bring a notable improvement of the results particularly for the moderate TSR: the explored angles of attack are indeed smaller than at low TSR.

Finally, we make clear that the production simulations of sections 3.2, 3.3 etc. do not include this correction: Simulations of these sections were run without the curvature correction investigated above (or equivalently, the simulated corresponds to a machine with a blade pitched outwards by $\alpha_0 = 1.65°$).

———————————

**Comment 4.** P7, the maximum angle of attack are not at $\theta = 90$ and $270$ degrees

—————

The maximum angle of attack are in the vicinity of $\theta = 90$ and $270$ degrees. "The tip vortices are the strongest in the vicinity of the upstream- and downstream-most positions of the blades (around $\theta = 90°$ and $270°$) where the blades operate at their maximum angle of attack."

———————————

**Comment 5.** P9, "At the design TSR, the blade exploits the delayed stall at its most:" – please explain what this means.

—————

What we meant is that in this regime (at the design TSR), there is an optimum between the delay on the circulation development and the occurence of airfoil (deep) stall. Here is the qualitative explanation of the trade off. Due to dynamic stall, the maximum angle of attack, circulation and thus normal force

occur after the passage of the maximum of geometric angle of attack ($\theta = 90°$ and $270°$), and this benefit is increasing as long as the TSR decreases. However, decreasing the TSR also means increasing the max value of the AoA. At some point, leading edge vortex shedding will occur (which is here beneficial), but the lower the TSR, the sooner it will appear, and so the shorter the delay before the circulation drops (after the leading edge vortex passage). This phenomenon can be observed in Figure 3b, which shows the normal force (and thus the circulation) drop dramatically just before $90°$. We note that the drop location is in agreement with the expected behavior of the Dynamic Stall model and parameters.

We consider the addition of the following clarification: "At the design TSR, the blade exploits the delayed stall at its most: its circulation keeps increasing, well past $\theta = 90°$, and then smoothly decreases. This is explained by two phenomena: (1) the airfoil experiences the highest delay in the circulation development (beneficial in this case as it widens the extent of torque production by the blade); (2) the leading edge vortex does not introduce a sharp drop in circulation yet (which clearly happens at lower TSR, see Fig.3b)."

———————————————

**Comment 6.** P9, "this mechanism, most visible in fig 4a, is well known in vortex dynamics and had already been identified on aircraft wakes." I suggest a figure where you highlight this event. It is a too complex process for a reader to follow form this short description.

——————

We propose the addition of the figure here labeled 2, to illustrate the vortex interactions.

———————————————

**Comment 7.** P9, "One needs to add additional terms to enforce an outflow conditions for this otherwise clipped vorticity field; the present study does precisely that by enforcing a normal outflow velocity through its Fourier-based

[Figure]

Figure 2: Vortex interactions are visible on the side of the wake, here illustrated in the area behind the bottom left corner of the VAWT, at successive times. The turbine is on the right, and the velocity is directed to the left.

solver.? This explanation is not clear. a. The comparison with the work of Scheurich uses TSR as a term of comparison; wouldn't loading be a more relevant term? The strength of the tip vortex is dependent also on the airfoil used; are these comparable? b. Other authors have used free wake vortex filament models, and have seen the same effect of inboard motion. How does this hold with the suggestion that a term should be added to the FFT solution in a meshed domain?"
* * *
This meant to say that, contrarily to other methods like free vortex filaments methods, the outflow conditions does not simply consist in clipping the vorticity. In our case, outflow boundary conditions are applied on the velocity field at the outflow plane. In the present case, these specific conditions are chosen so that the velocity field at the outflow plane is purely normal.

- We agree with the reviewer that the loading is a relevant parameter for comparison. Initially, we were expecting that, the loading being so closely

linked with TSR, we wouldn't be too far from Scheurich's loading by using the same TSR.

- The airfoil used is the same as for Scheurich's experiment. More precisely, we are using the very same Dynamic Stall model with the same coefficients. However, in the present simulations, the pitching angle was not adapted in order to take into account an effect of curvature of the relative flow (see answer to comment 3) and it is not clear whether Scheurich et al. are doing it too. This generates an uncertainty with respect to the loading of the blade as a function of the TSR, which points toward the Reviewer's previous remark.

- Our concern with respect to the work of Scheurich was that the size of his domain could be sufficiently small so that the clipping performed at the outflow could significantly influence the velocity field at the turbine. This was one possible explanation of the discrepancy between the velocity fields. With a longer domain and the presence of the normal outflow conditions, our simulation is more accurate in this respect.

  Withstanding the remark of the reviewer, though, other factors influencing the loading of the blade could explain the difference between the velocity fields.

Definitely, a new comparison based on a case at the same loading would be an improvement. The effect of curvature correction should also be sorted out.

―――――――――――――

**Comment 8.** P10, "The number of blades also has a strong influence; the two-bladed machine of Section 3.1 exhibits such vortex-blade collisions, in spite of its high loading." How does this relationship work?

―――――――

The purpose of this first study was not to investigate in details the influence of the number of turbine blades. However, it is clear that it plays a major role

in the geometry of the wake. To answer the Reviewer's comment, a detailed analysis of the interaction of the shed vortex sheet with the downstream blade and the different kinds of vortex interactions in this configuration should be performed.

————————————————

**Comment 9.** P11, For the discussion about figure 6, to get a better insight into the mechanism of turbulence creation in the wake corners, would it maybe be valuable to add an impression from the side as well? Then it is easier to see what happens in the corners.

——————————

Data required to build statistics in that plane were not collected. We may only refer to 3D views to get insights into the mechanisms of turbulence creation in the wake corners: we can then refer to the discussion above on vortex reconnections and also Fig.2 of the present document.

————————————————

**Comment 10.** P12,"The deformation of the velocity deficit clearly hints at the presence of mean streamwise vortices along the corners of the wake, a clear departure from a HAWT wake" This is only true for a HAWT subject to axial flow, but certainly not for inclined inflow for which a lateral force component is apparent (which is in practice always the case).

——————————

We totally agree with the reviewer and consider the following correction: "[...] a clear departure from a HAWT wake with no sideslip angle."

————————————————

**Comment 11.** P12, ". . ., which makes the use of a pointwise velocity deficit unsuitable." - suggestion -Which makes the definition of a velocity deficit profile based upon a single characteristic point unsuitable.

——————————

This Reviewer's comment makes the sentence clearer. We agree with the suggestion of correction, except that we consider to adapt the term "profile" which could be misleading in that context. Therefore, the following correction will be brought: "[...] which makes the definition of a velocity deficit evolution based upon a single characteristic point unsuitable."

———————————

**Comment 12.** P13, The value of S1, due to continuity equation, should always be zero, except for the fact that Uinf is corrected to the local velocity outside of the wake, which is larger than Uinf. Please explain the use of S1, and its modified application.

—————

The Reviewer's comment holds for a situation with no-through flow conditions imposed at some finite distance in the transverse directions, i.e. imposing a blockage. This would be the case for a wind tunnel test, and we agree that, in that case, S1 must asymptotically go to zero. However, in our simulations, there is a leakage flow through the boundary on the sides, which is permitted by the unbounded conditions of our velocity solver. Thus, the mass flow rate at the inflow of our simulation domain may be different from the mass flow rate at the outflow. Therefore, it is not that surprising that the value of S1 does not tend to 0 in our analyses, because of the transversal component of the velocity on the side boundaries. For the very same reason, the velocity which is recovered outside of the wake is exactly $U_\infty$ and must not be corrected.

Finally, the authors had introduced S1 in a parallel with a displacement thickness, as a means to measure how fast the wake decays in terms of velocity deficit.

———————————

**Comment 13.** P13, maybe add a reference for the $x/D \sim 50$ statement? And probably a discussion about the decay laws for HAWTS, in practice these seem

to deviate significantly from bluff-body flow rules in case of turbulent inflow.
* * *
References to that statement can be found in [5] (pp.151) and also in [6]. It is valid for bluff bodies, indeed. We agree that a thorough comparison with observed HAWT wake decay laws is an interesting avenue for future work. Indeed, as noted by the Reviewer, it was shown by [7] that, although the velocity deficit behind a HAWT without inflow turbulence behaves similarly to what's observed behind bluff bodies (i.e. a pointwise velocity deficit in $x^{-2/3}$), it's not the case with a turbulent inflow (where the bluff body theory would expect the pointwise velocity deficit to behave like $x^{-1}$).

To extend a bit our discussion on the asymptotic behavior of the wake (without going into details), we propose adding to the original text: "The decay observed for $x/D > 5$ for most of the configurations does however hint at a power-law-like behavior. For HAWTs, it has been observed that the decay deviates significantly from the bluff body behavior in the presence of a turbulence inflow [7]; similarly, it will be interesting to assess the sensitivity of VAWT wake decay with respect the turbulence intensity."

**References (local numbering)**

[1] D. Castelein, Dynamic stall on vertical axis wind turbines - creating a benchmark of vertical axis wind turbines in dynamic stall for validating numerical models, Master's thesis, Technische Universiteit Delft (2015).

[2] A. Bianchini, E. A. Carnevale, L. Ferrari, A model to account for the virtual camber effect in the performance prediction of an h-darrieus vawt using the momentum models, Wind Engineering 35 (4) (2011) 465–482. doi:10.1260/0309-524X.35.4.465.

[3] P. Migliore, W. Wolfe, Some effects of flow curvature on the performance of Darrieus wind turbines, American Institute of Aeronautics and Astronautics, 1979. doi:doi:10.2514/6.1979-112.

[4] L. Beaudet, Etude expérimentale et numérique du décrochage dynamique sur une éolienne à axe vertical de forte solidité, Ph.D. thesis, Université de Poitiers (2014).

[5] S. B. Pope, Turbulent flows, IOP Publishing, 2001.

[6] H. Tennekes, J. L. Lumley, A First Course in Turbulence, MIT Press, 1972.

[7] I. Litvinov, I. Naumov, V. Okulov, R. F. Mikkelsen, Comparison of far wakes behind a solid disk and a three-blade rotor, Journal of Flow Visualization and Image Processing 22 (4).

---

## Author Comment (AC2) · 4 Apr 2017

Dear reviewers, dear editor,

Please find attached the resulting modified manuscript.

We of course remain at your disposition if you have further questions.

For the authors,

Best regards,

P. Chatelain

[Figure]

Please also note the supplement to this comment:
http://www.wind-energ-sci-discuss.net/wes-2016-56/wes-2016-56-AC2-
supplement.pdf

---

## Author Response (AR2)

**Vortex Particle-Mesh simulations of Vertical Axis Wind Turbine flows: from the blade aerodynamics to the very far wake**
**Reply to editors**

P Chatelain[a], M Duponcheel[a], D-G Caprace[a], Yves Marichal[b], G Winckelmans[a]

[a]*Institute of Mechanics, Materials and Civil Engineering, Université catholique de Louvain, 1348 Louvain-la-Neuve, Belgium*
[b]*Wake Prediction Technologies (WaPT), Rue Louis de Geer 6, 1348 Louvain-la-Neuve, Belgium*
* * ** * *
We have addressed all the changes by the editors and reviewers. They are listed below.
* * *
**Comment 1.** I suggest you make the discussion of the definition of S1, and the outflow on the sides of the domain, clearer.
* * *
We have expanded the discussion around the definition of $S_1$: These diagnostics correspond to integrals of flux quantities in cross-stream sections located at a distance $x$ downstream of the turbine axis; their practical implementation approximates these integrals through quadrature over finite square sections $[-3\,D, 3\,D] \times [-3\,D, 3\,D]$. Because our Biot-Savart solver enforces transverse unbounded conditions exactly, it allows a transverse mass flow due to blockage. As a consequence, $S_1$, shown in Fig. **??**, does not vanish (as it would have for a solver with no-through flow boundaries); it quantifies the blockage effect caused by the wake on the flow."
* * *
**Comment 2.** Already in the abstract, please indicate the nature of the "unexpected topological flow features". It is neither perfectly clear from the conclusion what is "unexpected".
* * *
We agree with the editor that this was not clear. We have removed the adjective "unexpected", made the features more explicit, and thus changed the abstract.
* * *
**Comment 3.** Related to the previous point, does Figure 6 show the resolved TKE, or total TKE (resolved+modeled)?
* * *
Figure 6 shows resolved TKE only; this is now mentioned in the caption.
* * *
**Comment 4.** In addition to the comment by the associate editor, would it be possible to add to section 6 that if anyone is interested in (parts of) the data then please contact the authors?
* * *
Changed accordingly.